Evaluation of three wheat (Triticum aestivum L.) cultivars as sensitive Cd biomarkers during the seedling stage

He Chuntao 1 2
Ding Zhihai 2
Mubeen Samavia 2
Guo Xuying 2
Fu Huiling 2
Xin Guorong 1 2 lssxgr@mail.sysu.edu.cn
1 Guangdong Provincial Key Laboratory of Plant Resources, School of Agriculture, Sun Yat-sen Universtiy , Guangzhou , China
2 School of Life Sciences, Sun Yat-sen Universtiy , Guangzhou , China
Anderson Todd
Electronic publication date: 2020 Jan 27
Publication date: 2020
Volume: 8
Electronic Location ID: e8478
Received 2019 Oct 31; Accepted 2019 Dec 27
Copyright: © 2020 He et al.
Copyright year: 2020
Copyright holder: He et al.
License: This is an open access article distributed under the terms of the Creative Commons Attribution License, which permits unrestricted use, distribution, reproduction and adaptation in any medium and for any purpose provided that it is properly attributed. For attribution, the original author(s), title, publication source (PeerJ) and either DOI or URL of the article must be cited.
License URL: https://creativecommons.org/licenses/by/4.0/

Keywords: Cadmium, Triticum aestivum L., Root morphology, Physiological response, Cd sensitivity

Funding: National Water Pollution Control and Governance of Science and Technology Major Special 2009 ZX07528-001 Zhang-Hongda Science Foundation of Sun Yat-sen University This work was supported by the National Water Pollution Control and Governance of Science and Technology Major Special (2009 ZX07528-001) and the Zhang-Hongda Science Foundation of Sun Yat-sen University. The funders had no role in study design, data collection and analysis, decision to publish, or preparation of the manuscript.

==============================
Sensitive seedling crops have been developed to monitor Cadmium (Cd) contamination in agricultural soil. In the present study, 18 parameters involving growth conditions and physiological performances were assessed to evaluate Cd-responses of three wheat (Triticum aestivum L.) cultivars, Xihan1 (XH), Longzhong1 (LZ) and Dingfeng16 (DF). Principle component analysis illustrated that Factor 1, representing growth performance, soluble sugar content and catalase activity, responded to the Cd treatments in a dose dependent manner, while Factor 2 represented by chlorophyll content and germinating root growth was mainly dependent on cultivar differences. Higher inhibition rates were observed in growth performance than in physiological responses, with the highest inhibition rates of shoot biomasses (39.6%), root length (58.7%), root tip number (57.8%) and bifurcation number (83.2%), even under the lowest Cd treatment (2.5 mg·L−1). According to the Cd toxicity sensitivity evaluation, DF exerted highest tolerance to Cd stress in root growth while LZ was more sensitive to Cd stress, suggesting LZ as an ideal Cd contaminant biomarker. This study will provide novel insight into the cultivar-dependent response during using wheat seedlings as Cd biomarkers.

Introduction

Cadmium (Cd) contamination in agricultural soil raises the human health risk of Cd exposure through crop consumption. Numerous efforts were taken to alleviate the Cd contamination in crops such as the soil remediation (Gonzalez, Gil-Diaz & Lobo, 2017), agronomic management and low-Cd accumulating cultivar breeding (Huang et al., 2017). Nevertheless, the application of biomarker to predict the Cd contaminant in agricultural soil also plays an important role in ensuring the safe production of crops. The plant responses to Cd have been widely applied in environmental biomonitoring (Modlitbová et al., 2018).

Previous studies have focused on growth, oxidative stress, photosynthesis and other physiological alterations when exploring Cd-stress responses in plants. The growth repressions of shoots and roots in plants have been considered as direct symptoms of Cd toxicity (Liu et al., 2016; Zhan et al., 2017). Also, the oxidative stress (Haluskova et al., 2010; Yan et al., 2016) and adjustments in photosynthesis and energy metabolism (Elloumi et al., 2014; Ozfidan-Konakci et al., 2018) are apparent deficiencies caused by Cd stress in plant. However, to the best of our knowledge, the ideal sensitive biomarkers of Cd contamination have not been sufficiently explored.

Wheat (Triticum aestivum L.) is one of the most important crops in the world and serves as a staple food for over 50% global population (Rizwan et al., 2017; Rehman et al., 2018). Gonzalez, Gil-Diaz & Lobo (2017) explored Cd phytoremediation capacity in wheat and found out that the Cd tolerance of wheat cultivars was lower than that of barley cultivars, suggesting that the high Cd sensitivity of wheats might benefit Cd contamination monitoring. The responses of wheat seedlings to Cd suggested the possibility of their application as Cd contamination biomarkers (Gajewska et al., 2013; Ge et al., 2009). Additionally, Cd accumulations in durum wheat were significantly different among cultivars (Vergine et al., 2017), indicating that there are cultivar-dependent responses to Cd in wheat. However, few study has focused on the cultivar-dependent property of wheat seedlings as Cd bio-indicators. The present study aimed to evaluate the Cd sensitivity of three widely planted wheat cultivars in west-northern China and provide valuable information about the symptoms of wheats grown in Cd polluted soils.

One winter and two spring wheat cultivars were employed to examine growth deficiencies and the feasibility of using wheat seedlings as Cd biomarkers. Biomasses, root system morphological characteristics and the physiological responses of antioxidant and photosynthetic capacities were assessed to evaluate the Cd toxicity of different wheat cultivars. This study aimed to: (1) select efficient biomonitoring symptoms from the 18 tested parameters sensitive characteristics; (2) compare the cultivar-dependent Cd responses among three cultivars; and (3) evaluate the suitability of using these cultivars as biomarkers of Cd contamination according to their Cd sensitivities.

Materials and Methods

Plant materials

Three widely planted wheat cultivars were used in this study. The wheat cultivars Xihan1 (XH) and Longzhong1 (LZ) were provided by the Agronomy College, Gansu Agricultural University. XH (spring wheat) and LZ (winter wheat) were widely cultivated local cultivars planted in spring and winter, respectively. Dingfeng16 (DF), a newly bred spring wheat cultivar with high drought and disease resistances, was kindly provided by the Dry Farming Scientific Research Extension Center of Dingxi City, Gansu Province.

Culture and treatments

The seeds of the three cultivars were washed with ultrapure water three times after sterilisation in 1% NaClO for 20 min. Seed germination was carried out in 9 cm plates with exactly 20 seeds placed in each plate. Germinating seeds were exposed to different Cd concentrations (2.5 mg·L−1, 5 mg·L−1, 10 mg·L−1, 20 mg·L−1 and 40 mg·L−1) and seeds in ultrapure water were set as control (CK). Each treatment was performed in three replicates. The plates were then kept at 25 °C in a plant incubator. After 36 h of germination, the longest and total root lengths of each seed were measured.

The germinated seedlings were transferred into culture flasks containing Hoagland nutrient solution and were kept in light/dark conditions at 25 °C/18 °C for 16 h/8 h. The treatment of Cd (in CdCl2) continued according to seedling germination: 2.5 mg·L−1, 5 mg·L−1, 10 mg·L−1, 20 mg·L−1 and 40 mg·L−1 and the seedlings grown in the Hoagland nutrient solution were set as control. Three seedlings were planted in each flask and each treatment was performed in triplicate. The roots and shoots were harvested after 7 days.

Determination of root growth condition

The root and leaf images were obtained using a scanner (CanoScan LiDe 00F) in 300 dpi. Measurements of root length, surface area, volume, diameter, root tip number, bifurcate number and leaf length were accomplished using Wseen’s Universal Plant Image Analysis System. The longest root lengths and the summary of all measured root systems (the total root length) of germinated and seedling plants were recorded respectively.

Assessment of the physiological index

Physiological index assessments were carried out according to previously reported methods. Measurements were performed following the instructions for root activity (Man et al., 2016), catalase activity (Li et al., 2013), soluble sugar content (Verma & Dubey, 2001), carotenoid content (Panda, Chaudhury & Khan, 2003) and chlorophyll contents (Feng, Zhu & Li, 2013), respectively.

Calculation of toxicity sensitivity

The inhibition rate (IR) was adopted to evaluate Cd toxicity as an independent parameter, with the lower IR representing a lower toxicity sensitivity and the calculation formulation as follows: IR=1−Ytreatment/Yck

where Ytreatment represents the value under treatment and Yck represents the value under control.

The toxicity sensitivity (TS) of each tested indexes among three cultivars under different treatments was calculated according to the following formulation: TSij∧=(IRij−IRjmin)/(IRjmax−IRjmin)

where IRij is the IR of cultivar i and Xjmin and Xjmax are the minimum and maximum IRs among the three cultivars under Cd treatment. The TS of tested cultivars were calculated by different Cd treatments. And the TS of every Cd treatments was represented by the average TS∧ij of the 18 tested indexes for three cultivars, respectively.

Statistical analysis

The statistical analysis was carried out using IBM SPSS Statistics 22 (SPSS Inc., Chicago, IL, USA). One-way analysis of variance (ANOVA) was applied to compare the performances among different Cd treatments within the same cultivar and the difference among the three wheat cultivars under the same treatments. The LSD test was adopted in the ANOVA analysis with significance at p < 0.05. Principal component analysis was performed to show the relationship among the three tested cultivars and the tested indexes under Cd stress. The p value ≤ 0.05 was considered statistically significant.

Results

Root lengths under Cd treatments during germination

The root lengths of the three wheat cultivars under different Cd treatments during seed germination are shown in Table 1. Slight increases in root length during germination were found under the lower Cd concentration treatments. The longest root length during germination was found at 2.5 mg·L−1 Cd treatment for XH and at 5 mg·L−1 Cd treatment for LZ and DF. About 30% promotion was found in root length at 5 mg·L−1 for the cultivar DF when compared to CK. High Cd treatments repressed the root lengths in all three wheat cultivars. Under the 40 mg·L−1 Cd treatment, the IRs reached 27.4%, 42.5% and 20.2% for XH, LZ and DF, respectively. Similar trends were found for the longest root lengths of three cultivars under Cd treatments with the highest IRs of 41.1%, 45.7% and 27.8% for XH, LZ and DF at 40 mg·L−1 Cd treatment.

Table 1 The root growth performance of different wheat cultivars under Cd treatments.

The root growth status during germinating and seedling were shown in three cultivars under different Cd stresses. The different letters represent significant difference among different Cd treatment concentrations within the same cultivar.

Cd treatments (mg·L−1)	Germinating	Seedling	
		Total length (cm)	Total length (cm)	Area surface (cm2)	Volume (cm3)	Longest length (cm)	Tips	Bifurcation	Diameter (mm)	
XH	0	5.18 ± 0.0.43ab	70.71 ± 21.27a	7.66 ± 3.19a	2.48 ± 2.59a	13.28 ± 0.62a	19.89 ± 4.30a	15.89 ± 3.67a	0.51 ± 0.08b	
2.5	5.71 ± 0.20a	30.21 ± 4.45b	4.74 ± 0.83a	0.20 ± 0.17b	6.86 ± 1.25b	11.00 ± 3.21b	7.89 ± 1.35b	0.64 ± 0.16ab	
5	5.09 ± 0.34ab	27.43 ± 5.39b	6.62 ± 1.57a	0.15 ± 0.05b	8.50 ± 1.01b	6.89 ± 0.69c	2.28 ± 0.25c	0.82 ± 0.03a	
10	5.21 ± 0.46ab	20.61 ± 3.85bc	3.54 ± 1.65b	0.07 ± 0.03bc	5.94 ± 0.82c	5.44 ± 0.51c	1.50 ± 0.5c	0.70 ± 0.01b	
20	4.61 ± 0.32b	15.24 ± 3.64bc	3.01 ± 1.07bc	0.05 ± 0.01c	5.69 ± 1.12c	4.33 ± 0.58c	0.00 ± 0.00d	0.60 ± 0.06b	
40	3.76 ± 0.18c	8.49 ± 0.93c	1.28 ± 0.55c	0.02 ± 0.01c	2.96 ± 0.22d	3.44 ± 0.19c	0.00 ± 0.00d	0.57 ± 0.13b	
LZ	0	4.89 ± 0.54a	61.26 ± 8.75a	6.66 ± 1.22a	0.69 ± 0.57a	13.56 ± 0.97a	15.67 ± 2.31a	12.89 ± 1.95a	0.48 ± 0.03b	
2.5	4.99 ± 1.32a	25.32 ± 6.68b	3.63 ± 0.83b	0.15 ± 0.15b	9.66 ± 1.39b	6.61 ± 1.08b	2.17 ± 2.02b	0.45 ± 0.12b	
5	5.61 ± 0.79a	18.73 ± 4.58bc	3.51 ± 0.73b	0.07 ± 0.02b	6.48 ± 2.17c	7.33 ± 2.08b	2.22 ± 2.11b	0.66 ± 0.04a	
10	4.82 ± 0.32a	16.57 ± 2.13bc	3.01 ± 0.32bc	0.05 ± 0.01bc	4.94 ± 1.02cd	5.11 ± 0.19bc	0.00 ± 0.00c	0.65 ± 0.04a	
20	4.35 ± 0.48a	12.57 ± 4.15cd	1.81 ± 0.54c	0.03 ± 0.01cd	3.67 ± 0.82de	4.67 ± 0.67c	0.00 ± 0.00c	0.59 ± 0.06ab	
40	2.81 ± 0.19b	3.88 ± 0.69d	0.56 ± 0.16d	0.01 ± 0.00d	1.75 ± 0.41e	3.00 ± 0.00c	0.00 ± 0.00c	0.61 ± 0.06ab	
DF	0	3.42 ± 0.12c	55.85 ± 19.42a	6.94 ± 2.64a	0.73 ± 0.57a	9.39 ± 1.54a	24.33 ± 9.75a	17.83 ± 9.93a	0.53 ± 0.08b	
2.5	4.21 ± 0.25ab	41.24 ± 19.65b	4.20 ± 0.89b	0.13 ± 0.12ab	8.46 ± 1.82a	14.67 ± 5.17b	8.56 ± 4.44b	0.51 ± 0.05b	
5	4.62 ± 0.21a	34.36 ± 5.61b	7.15 ± 0.88b	0.18 ± 0.02ab	8.57 ± 1.36ab	12.00 ± 1.73bc	6.67 ± 2.03bc	0.76 ± 0.03a	
10	3.80 ± 0.32bc	20.11 ± 1.27c	4.36 ± 1.11b	0.09 ± 0.03bc	6.02 ± 0.76b	7.78 ± 0.51cd	5.00 ± 1.00cd	0.68 ± 0.08a	
20	3.28 ± 0.41cd	12.31 ± 1.54cd	1.89 ± 0.50c	0.04 ± 0.02c	3.01 ± 0.21bc	5.78 ± 0.50d	1.00 ± 1.00d	0.68 ± 0.02ab	
40	2.72 ± 0.16d	5.02 ± 0.73d	0.67 ± 0.07d	0.01 ± 0.00c	1.68 ± 0.11c	4.11 ± 0.84d	0.67 ± 1.15d	0.50 ± 0.01b	

Seedling biomasses under Cd treatments

The root and shoot biomasses under different Cd treatments are shown in Fig. 1. The root biomass was highest in XH among the three cultivars under different Cd treatments. The Cd treatments caused slight increases of the root biomasses of XH and DF cultivars under 0.5 and 5 mg·L−1 Cd treatment compared to CK (p < 0.05). The highest IRs for XH and DF reached 66.1% and 78.1%, respectively, at 40 mg·L−1 Cd concentration. Notably, the significant inhibition of root biomass growth was found in XH and DF under 10 mg·L−1 Cd treatment, while significant repression was observed for the root biomass of LZ only under the 40 mg·L−1 Cd treatment (p < 0.05).

Figure 1 The biomasses of shoot and root for three wheat cultivars under different Cd treatments.

The different lowercase letters represented significant difference was found among different Cd treatments in shoot or root biomasses for the same wheat cultivar (p < 0.05).

The biomasses of the shoots under Cd treatments were similar across different cultivars, showing a decrease as Cd concentrations increased from the 10 mg·L−1. Significant decreases in shoot biomasses were first observed during the 2.5 mg·L−1 treatment for XH and LZ which was observed after Cd treatments reached over 20 mg·L−1 in DF. The highest IRs in shoot biomasses were 76.0%, 81.0% and 79.7% for XH, LZ and DF under 40 mg·L−1 treatment, respectively.

Root morphology under Cd treatments

All the tested root growth parameters decreased with increasing Cd concentrations, except for root diameters (Table 1). For the three wheat cultivars, a significant decrease in total root length was found under different Cd treatments when compared to the CK (p < 0.05) and the total root lengths decreased with increasing Cd concentrations. Under the 40 mg·L−1 Cd treatment, the repression rates of root lengths of XH, LZ and DF have reached to 93.6%.

For the three cultivars, the Cd treatments significantly reduced the root surface areas (p < 0.05). The highest IRs reached 83.3%, 91.6% and 90.3% under 40 mg·L−1 Cd treatment for XH, LZ and DF, respectively. Similar decreasing trends were found for the root volumes as Cd concentrations increased. The IRs of root volumes under the 40 mg·L−1 Cd treatment were 92.8%, 97.3% and 96.2% for XH, LZ and DF, respectively.

Unlike root length and surface area, the root average diameters of the three cultivars did not show significant changes with increased Cd concentrations. The highest average root diameters in the three cultivars (XH: 0.82 mm, LZ: 0.66 mm, DF: 0.76 mm) were found under the 5 mg·L−1 Cd treatment, followed by the 10 mg·L−1 Cd treatment, both significantly higher than the CK (p < 0.05). The rates of increase under the 5 mg·L−1 Cd treatment in XH, LZ and DF were 61.7%, 25.9% and 42.9%, respectively.

The root system bifurcation and tip numbers decreased significantly under Cd treatments when compared to CK (p < 0.05) in the three cultivars and the inhibition levels increased as Cd concentrations increased. There was no bifurcation in the roots of XH and LZ with IRs of 100% under 20 and 40 mg·L−1 Cd treatments. Similar notable inhibitions were found in the root tip numbers for the three cultivars under different Cd treatments.

The physiological responses under Cd treatments

The root activity of XH increased significantly under the 5 mg·L−1 Cd treatment (promotion rate: 124%) while the highest root activity of LZ was seen under the 10 mg·L−1 Cd treatment (promotion rate: 119%) (Table 2). The root activities of DF remained stable under different Cd treatments until the Cd concentration reached 40 mg·L−1, where a significant reduction occurred compared to the CK (IR: 37.8%). The catalase activities of leaves increased along with increasing Cd concentrations. The highest elevation rates were 525%, 232% and 198% for XH (40 mg·L−1), LZ (40 mg·L−1) and DF (20 mg·L−1, the data for 40 mg·L−1 is missing), respectively. Overall, the soluble sugar content under different Cd treatments was significantly higher than that of CK. The soluble sugar contents were significantly induced in the leaves of tested cultivars under Cd treatments (p < 0.05). The highest soluble sugar contents were 4.69%, 5.04% and 4.71% for XH, LZ and DF, under 2.5 mg·L−1 Cd treatment, with promotion rates of 347%, 413% and 306%, respectively. However, a decrease in soluble sugar content was observed with increasing Cd treatment. The chlorophyll a contents of DF were reduced significantly under Cd treatments over 10 mg·L−1 and the IR reached 32.4%. The Cd treatments showed no influence on the chlorophyll a contents of XH. For LZ, significant reductions of the chlorophyll a contents were observed only under the 40 mg·L−1 treatment. The ratio of chlorophyll a to chlorophyll b content reduced significantly under the 40 mg·L−1 Cd treatment.

Table 2 The physiological responses during seedling were shown in three cultivars under different Cd stresses.

All the tested indexes were measured based on the fresh weight. The different letters represent significant difference among different Cd treatment concentrations within the same cultivar.

Cd treatments (mg·L−1)	Catalase activity (mg·g−1·min−1)	Soluble sugar content (%)	Root activity TTC oxidative (μg·g−1·h−1)	Chlorophyll a (mg·g−1)	Chlorophyll b (mg·g−1)	Carotenoid (mg·g−1)	Ratio of chlorophyll a to chlorophyll b	
XH	0	8.82 ± 2.41c	1.05 ± 0.14d	2,370 ± 604b	1.12 ± 0.09a	0.16 ± 0.04a	0.28 ± 0.05a	6.89 ± 0.66a	
2.5	18.87 ± 4.63c	4.69 ± 0.08a	3,985 ± 571ab	1.03 ± 0.38a	0.15 ± 0.02a	0.29 ± 0.02a	6.82 ± 0.56a	
5	13.37 ± 3.41c	4.11 ± 0.02ab	5,314 ± 1171a	0.94 ± 0.16a	0.17 ± 0.04a	0.25 ± 0.02a	5.79 ± 1.23ab	
10	18.42 ± 3.91c	3.98 ± 0.06b	3,213 ± 1063abc	1.06 ± 0.29a	0.18 ± 0.01a	0.27 ± 0.02a	5.97 ± 0.70ab	
20	36.94 ± 15.92b	3.00 ± 0.16c	3,468 ± 2164abc	1.00 ± 0.36a	0.17 ± 0.03a	0.25 ± 0.03a	5.95 ± 0.76ab	
40	55.11 ± 0.48a	3.20 ± 0.28c	1,642 ± 599c	0.84 ± 2.46a	0.16 ± 0.12a	0.19 ± 0.13b	5.47 ± 0.70b	
LZ	0	17.73 ± 2.57c	0.98 ± 0.04c	1,614 ± 339b	1.28 ± 0.06ab	0.22 ± 0.02a	0.22 ± 0.19ab	5.85 ± 0.58ab	
2.5	21.33 ± 5.23c	5.04 ± 0.24a	2,592 ± 1675ab	1.43 ± 0.30a	0.21 ± 0.05a	0.40 ± 0.06a	7.00 ± 0.40a	
5	23.80 ± 7.41c	4.36 ± 0.07ab	3,207 ± 1339ab	1.12 ± 0.41ab	0.20 ± 0.03a	0.29 ± 0.01ab	5.67 ± 0.45b	
10	26.84 ± 3.79b	4.23 ± 0.03ab	3,542 ± 814a	1.08 ± 0.58ab	0.21 ± 0.01a	0.27 ± 0.01b	5.10 ± 0.10b	
20	40.57 ± 4.35b	4.06 ± 0.08b	1,888 ± 999ab	1.14 ± 0.35ab	0.18 ± 0.00a	0.29 ± 0.01b	6.32 ± 0.46ab	
40	58.80 ± 7.63a	3.76 ± 0.14b	1,590 ± 486b	1.01 ± 0.91b	0.20 ± 0.02a	0.23 ± 0.07b	4.95 ± 1.17b	
DF	0	10.55 ± 6.70b	1.16 ± 0.09c	3,907 ± 1432a	1.06 ± 0.16a	0.18 ± 0.03a	0.29 ± 0.02a	6.12 ± 0.57b	
2.5	12.85 ± 1.76b	4.71 ± 0.12a	2,466 ± 412ab	0.85 ± 1.12ab	0.11 ± 0.02a	0.25 ± 0.08ab	7.80 ± 0.76a	
5	16.52 ± 4.03b	3.98 ± 0.05ab	1,807 ± 226b	0.95 ± 0.73abc	0.16 ± 0.02a	0.30 ± 0.01a	5.86 ± 0.93b	
10	28.14 ± 11.09ab	3.86 ± 0.03ab	1,978 ± 428ab	0.91 ± 0.77bc	0.14 ± 0.01a	0.25 ± 0.02ab	6.48 ± 0.42ab	
20	31.42 ± 5.54a	3.04 ± 0.07b	1,972 ± 627ab	0.82 ± 0.43bc	0.15 ± 0.03a	0.21 ± 0.02b	5.56 ± 0.83b	
40	–	3.12 ± 0.04b	2,468 ± 565ab	0.72 ± 3.41c	0.14 ± 0.00a	0.09 ± 0.08b	5.18 ± 0.29b	

The principal component analysis of responsive parameters of the three cultivars under Cd treatments

The loading plots of the responsive parameters of the three wheat cultivars are shown in Fig. 2. The total variance of the first two major principal components accounted for 67.5%, with 52.0% and 15.5% for Factor 1 and Factor 2, respectively. The factor score plot (Fig. 2A) presents the loading information of the sample distribution for both factors. The control samples and lower Cd treatment samples displayed higher scores in Factor 1. The highest scores were observed from the CK of the three wheat cultivars while the samples from higher Cd treatments shared the lowest scores in Factor 1. Unlike Factor 1, Factor 2 showed the cultivar-specific distribution patterns of the three wheat cultivar samples. The highest score of Factor 2 was occupied by samples from LZ (the winter wheat cultivar), while XH and especially DF were mainly observed in the lower scores of Factor 2.

Figure 2 The principle component analysis of the tested physiological growth parameters for three wheat cultivars.

(A) The loading scores of the tested samples on the principle components. Different Cd treatments were represented by ⚫ (control), + (2.5 mg·L−1), (5 mg·L−1), ◼ (10 mg·L−1), ♦ (20 mg·L−1) and ▴ (40 mg·L−1) and (B) the component matrix of the tested factor, each component explained 52.0% and 15.5% of the total variance.

Different responsive parameters were categorised according to their loadings on both factors (Fig. 2B). The responsive factors of growth performances such as root biomass, length, volume, surface area, tip/bifurcation numbers and leaf biomass were loaded in the high score of Factor 1. Conversely, soluble sugar and catalase contents were located in the low score of Factor 1. Chlorophyll content and germinating root lengths were mainly clustered in the high score of Factor 2.

The toxicity sensitivity of the three cultivars under Cd treatments

The TS of the three cultivars under different Cd treatments are shown in Table 3. The highest TS in the physiological response was found in DF, which exerted a higher Cd tolerance in root growth performance. The physiological response was mainly represented by the shoot response, with XH showing the most tolerance to Cd. Among all tested indexes, LZ displayed the highest TS under the lower Cd treatments of 2.5–10 mg·L−1, while DF showed higher sensitivity under the 20 and 40 mg·L−1 Cd treatments.

Table 3 The toxicity sensitivity of three cultivars under Cd treatments.

Cd treatments (mg·L−1)	Growth performance	Physiological response	Summary	
XH	LZ	DF	XH	LZ	DF	XH	LZ	DF	
2.5	0.491	0.862	0.149	0.438	0.102	0.857	0.472	0.582	0.410	
5	0.634	0.767	0.212	0.571	0.254	0.617	0.611	0.578	0.361	
10	0.602	0.644	0.339	0.466	0.466	0.774	0.552	0.578	0.499	
20	0.644	0.540	0.415	0.272	0.296	0.945	0.507	0.450	0.611	
40	0.391	0.667	0.566	0.480	0.044	0.859	0.425	0.425	0.680	

Discussion

The oxidative damages in tested wheat cultivars under Cd exposure

In the present study, chlorophyll content, carotenoid content and root activity in wheat were not sensitive to Cd according to their slight inhibitions even under the Cd concentration of 40 mg·L−1. Cd exposure has been reported to reduce chlorophyll a and particularly chlorophyll b content in Sinapisalba L. seedlings (Fargasova, 2001), which was different from the stable chlorophyll b content found in the tested cultivars in the current study. It has also been reported that photosynthetic parameters are greatly reduced in Cd-treated Brassica napus L. (Jhanji et al., 2012). Slight inhibitions of chlorophyll a in wheat cultivars tested under the 40 mg·L−1 Cd treatment and the ratio of chlorophyll a to chlorophyll b implied the deficiency of the photosynthesis system under high concentrations of Cd stress, which was considered as non-sensitive symptoms in the tested wheat cultivars. Unlike chlorophyll content, the dramatically increases even under lower Cd treatments (2.5 mg·L−1) in catalase activity and soluble sugar content suggested the oxidative status was one of the dominant responsive symptoms to Cd stress in wheat. Oxidative stress caused by Cd exposure has been widely reported in plants and catalase activity is a critical component in eliminating reactive oxygen species (Rady & Hemida, 2015). The catalase activity in the wheat cultivars exhibited a dose-dependent increase along with increasing Cd treatments, indicating that the higher oxidative stress caused by Cd is a sensitive symptom. Similarly, the alteration in soluble sugar content under Cd exposure was notable. Previous studies have shown that plants tend to accumulate extra soluble sugar under Cd stress (Marzban et al., 2017; Verma & Dubey, 2001). Soluble sugar is not only an important energy resource in plants, but is also a critical regulator in plant growth and oxidative status. The notable alterations of catalase activity and soluble sugar content indicate that these two physiological responses are potential biomarkers of Cd toxicity.

The ubiquitous growth inhibitions in tested wheat cultivars under Cd exposure

Compared to physiological responses, remarkable inhibitions were found in most of the growth parameters of the tested wheat cultivars under Cd stress. The Cd induced-growth inhibition is probably ascribed to the programmed cell death through endoplasmic reticulum (ER) stress. It was demonstrated that the Cd stress induced programmed cell death in Arabidopsis seedlings through the ER stress signalling (Xu et al., 2013). Besides, the ER stress was also responsible to the glucose 2 regulated protein, which might further influence the soluble sugar content in the present study. It has been previously reported that the Cd toxicity ranked root elongation > shoot elongation > germination rate in wheat (Chen et al., 2010). Our findings were consistent with this pattern where the most sensitive growth symptom of Cd is the root length, followed by the leaf length. Therefore, the tested wheat cultivars in the current study showed better Cd tolerance during the germinating stage than the seedling stage. Therefore, the growth parameters of shoot biomass and root length can be considered ideal and direct biomarkers of Cd exposure.

The growth performance of roots is more vulnerable to Cd exposure. Root growth inhibition and radial root swelling are characteristic symptoms in barley root tips after exposure to Cd (Liptakova et al., 2013; Lux et al., 2011). Since no significant change in average diameters was observed, it was supposed that the dominant Cd toxicity in the root system mainly relied on the repression of the elongation progress. The notable inhibitions of the surface area and root volume occurred along with the reduced root length under Cd treatments. The repressed root growth might be ascribed to the following explains. Firstly, the root elongation may have been blocked in the elongation zone due to oxidative stress. The early exposure of roots to Cd significantly increases the production of reactive oxygen species in the root’s proximal elongation zone, which could be alleviated by the NO donor (Alemayehu et al., 2015). Also, the boosted NO and O2− production is required for Cd-induced programmed cell death that is in turn induced by Cd in Lupinus luteus L. roots (Arasimowicz-Jelonek et al., 2012). Cd accumulation induces oxidative injury and cell death in the root tips of Brassica rapa (Lv et al., 2017), suggesting that Cd exposure would alter root growth through oxidative stress alteration. Secondly, the damaged root tip restricts the further root elongation under Cd exposure was supposed to be another important reason for the root growth inhibition. The elongation zone cells in the root tips are the very first accumulating target of the Cd ion (Shi et al., 2016b). Energy dispersive X-ray analysis suggests that Cd ions are localised in the meristem, elongation and mature zones in the root tips of Hordeum vulgare (Shi et al., 2016a). The tips are active in the growth, elongation, absorption and differentiation in roots. Structural damage to root tips has been observed for wheat cultivars grown in 100 μg Cd L−1 (Rizvi & Khan, 2017), which might be a possible explanation for the reduction in tip number and further inhibition of plant growth under Cd treatment. Also, the reduced bifurcation and tip numbers might be ascribed to the inflated hair zones of the root. Thirdly, the altered plant hormones in the root system may have inhibited the root growth. The bifurcation number represents the generation of lateral root, which is mediated by several plant hormones (Elobeid et al., 2012). Auxin was the most dominant hormone promoting lateral root formation, while cytokinin and abscisic acid suppressed lateral root formation (Tamás et al., 2012). The cell cycle of root tips was dramatically influenced by auxin homeostasis during Cd stress in various plants such as Sorghum bicolor seedlings (Zhan et al., 2017), Arabidopsis seedlings (Hu et al., 2013) and rice seedlings (Zhao et al., 2011, 2012).

Nonetheless, root diameter was the only root growth parameter that was not influenced by Cd exposure. Excessive Cd stress significantly increased the root average diameters in peanuts (Lu et al., 2013), soybean (Wang et al., 2016) and winter wheat (Qin et al., 2018). Increased root diameter under Cd stress is interpreted as compensatory growth and might be the consequence of altered root development involving xylogenesis, premature endodermis differentiation and lignification of cortical and stelar tissues. Thicker roots could act as a barrier protecting roots from metals (Bochicchio et al., 2015). Although root diameter is not an ideal Cd sensitive biomarker, it has provided information on the Cd tolerance of the tested cultivars.

Cultivar dependent Cd sensitivity in tested wheat cultivars under Cd exposure

Cultivar-dependent differential Cd tolerance has been reported in various plants such as maize (Akhtar et al., 2017), Amaranthus gangeticus (He et al., 2018), Italian ryegrass (Fang et al., 2017) and rice (Hou et al., 2018). In current study, different Cd tolerance and tissue vulnerability to Cd toxicity in three tested cultivars were unravelled. PCA analysis revealed that three cultivars performed differently in Factor 2, which mainly consisted of chloroplast contents and germinating growth performance. Although the chlorophyll contents were stable across different Cd treatments, higher chlorophyll content was found in LZ, which was the only winter wheat in this study. What’s more, the chlorophyll and carotenoid contents in LZ leaves increased during the 5 mg·L−1 Cd treatment, suggested higher Cd tolerance in photosynthesis system in LZ. Differently, higher Cd sensitivities in physiological responses indicated the Cd vulnerability on the photosynthesis and oxidative systems in the XH and DF. Hence, the different responsive mechanisms were developed in spring and winter wheat cultivars.

Although the shoot growth of the winter wheat cultivar LZ was more tolerant than that of the other two spring cultivars, the breeding cultivar DF displayed better root growth performance. The highest promotion rate at 5 mg·L−1 in germinating root length indicated that DF possessed a higher tolerance to Cd. These results suggest that DF has better Cd tolerance during germination. Additionally, a higher tolerance in DF was also observed according to its root growth conditions during seedling which might provide better absorption and transportation conditions for the shoots. Specially, the bifurcation number indicated that the root system of DF exhibited a higher tolerance to Cd stress than the other two cultivars, as the bifurcation number of LZ was reduced to zero under Cd concentrations over 10 mg·L−1, suggested a lower Cd toxicity and higher tolerance of the root growth system. The DF is a breeding cultivar with higher resistance to disease and environmental stress such as dryness, which might facilitate it higher Cd tolerance and lower TS in growth status under Cd stress.

The Cd TS evaluation of the three cultivars as biomarkers of Cd exposure was also in line with the PCA analysis. For the three tested wheat cultivars, LZ was the most vulnerable to Cd exposure in regarding to the 18 tested indexes. Although DF exhibited the highest sensitivity to the Cd induced photosynthesis and oxidative damage, the sensitive growth status alteration in LZ is a more direct symptom under Cd stress under different Cd concentration treatments, indicating that LZ is an ideal biomarker of Cd contamination. Our findings demonstrated the cultivar-dependent difference was a critical factor in selecting the Cd contaminated soil biomarkers, which would be of benefit in selection of biomarkers and phytoremediation to Cd contaminations.

Conclusion

This study has compared the Cd responsive performance in three wheat cultivars. Among three tested cultivars, growth status are more sensitive and direct symptoms than physiological response under Cd exposure, especially for the root morphology. The inhibition of root growth was probably caused by the repression of the elongation process and tip damage. The novel breeding cultivar DF showed a higher tolerance to Cd stress in its root growth. Ultimately, the winter wheat LZ was most sensitive to Cd stress and should be considered as an ideal biomarker of Cd contamination in soil.

Supplemental Information

Supplemental Information 1 Raw data.

Click here for additional data file.

Additional Information and Declarations

Competing Interests

Author Contributions

Data Availability

The authors declare that they have no competing interests.

Chuntao He conceived and designed the experiments, performed the experiments, analysed the data, prepared figures and/or tables, authored or reviewed drafts of the paper, and approved the final draft.

Zhihai Ding conceived and designed the experiments, prepared figures and/or tables, and approved the final draft.

Samavia Mubeen conceived and designed the experiments, prepared figures and/or tables, authored or reviewed drafts of the paper, and approved the final draft.

Xuying Guo conceived and designed the experiments, prepared figures and/or tables, and approved the final draft.

Huiling Fu analysed the data, prepared figures and/or tables, and approved the final draft.

Guorong Xin conceived and designed the experiments, authored or reviewed drafts of the paper, and approved the final draft.

The following information was supplied regarding data availability:

The raw measurements are available in the Supplemental Files.

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
