# Peer review of "Evaluation of three wheat (Triticum aestivum L.) cultivars as sensitive Cd biomarkers during the seedling stage"

_PeerJ, doi:10.7717/peerj.8478_

## Round 0.1 · original submission · Major Revisions

Please follow or respond to reviewer comments. When you have completed your revisions, I would encourage you to edit your manuscript to improve it's readability and impact.

Reviewer 1 ·

Basic reporting

With the aggravation of industrialization and urbanization, Cd pollutant has become a crucial abiotic stress. The MS (41895), using wheat seedlings as Cd biomarkers, evaluated the sensitivity of three wheat cultivars to Cd stress, and found that DF showed a higher tolerance, while LZ was a sensitive cultivar. These are of great importance for understanding the mechanism of Cd tolerance.

Experimental design

1.Which method was applied to compare the tested index in ANOVA? Ducan test, SNK test, or other.
2.The determination method of carotenoids was not presented in METHODS. Please check it.

Validity of the findings

1.Soluble sugar not only osmolyte and energy resource, but also reactive oxygen species scavenger, the latter should be discussed in DISCUSSION.
2.The growth performance (total length, volume, longest length, etc.) and physiological (CAT, soluble sugar, TTC, etc.) were inconsistent with the tolerance of wheat cultivars to Cd, which should be explained in DISCUSSION.

Additional comments

1.Language should be carefully revised, such as chlorophyll β, may be chlorophyll b.

Reviewer 2 ·

Basic reporting

The manuscript entitled: "Evaluation of three wheat (Triticum aestivum L.) cultivars as sensitive Cd biomarkers during the seedling stage" showed us the different manifestation in growth performance and physiological response to Cd exposure in wheat and assess the potential biomarker for Cd exposure. The contents fit the scope of the journal.

Experimental design

no comment

Validity of the findings

no comment

Additional comments

The manuscript entitled: "Evaluation of three wheat (Triticum aestivum L.) cultivars as sensitive Cd biomarkers during the seedling stage" showed us the different manifestation in growth performance and physiological response to Cd exposure in wheat and assess the potential biomarker for Cd exposure. The contents fit the scope of the journal. However, there are some open questions to need addressing.
Major comments
1. More research background and research purpose should be presented in Introduction.
2. Results should be more concise and accurate. The key data patterns should be highlighted.
3. The discussion should be more structured and focused on the reason of responsing Cd stress in wheat.
4. Conclusion should be carefully drawn. In this paper, only three cultivars of wheat were used to evaluate, however, some sentences in conclusion were not completely express this meaning.
Minor comments
Line 99-104 Theses sentences are not accurate. Please rewrite and present the clear treatment.
Line 119-120 this sentence is not clear. What dose mean “about 30% more significant growth”?
Line 146-147 what is objectives of comparison?
Line 177-178 It is hard to understand the presentation of sugar content.
Line 199-200 I can’t find root weight and leaf weight description in Figure 2B.
In table section, some indexes have no unit. Is the content of chlorophyll calculated based on dry weight or fresh weight? In table 2, the unit of root activity is “µ”rather than “u”.
In reference section, some journals abbreviation was not accurate, please check the format.

Reviewer 3 ·

Basic reporting

The article is written well.

Experimental design

Experimental design is correct.

Validity of the findings

the data is enough to support conclusions.

Additional comments

This paper describes a research about the responses of three wheat cultivars when exposed to Cd. The problem could be important, and currently worth to be considered. The manuscript has to improve in results and discussion chapter. So, I think this manuscript should be published in peer J after major revised.
1. Why do author select three wheat cultivars from province Gansu? Are they typical?
2. The abstract is no clear, please revise it.
3. Please explain why author select eighteen parameters of wheat during the seedling stage?
4. Please revise tables.
5. Please forecast practical significance of the results.
6. How reliable are the results of wheat seedlings as Cd biomarkers?

---

## Round 0.2 · accepted · Accept

Thank you for your efforts in improving your manuscript following reviewer comments.

Reviewer 3 ·

Basic reporting

IT IS OK.

Experimental design

IT IS OK.

Validity of the findings

IT IS OK.

Additional comments

Accept.